# Nutritional and Technological Optimization of Wheat-Chickpea- Milk Powder Composite Flour and Its Impact on Rheological and Sensorial Properties of Leavened Flat Bread

**DOI:** 10.3390/foods10081843

**Published:** 2021-08-10

**Authors:** Aouatif Benali, Youness En-nahli, Younès Noutfia, Adil Elbaouchi, Mohammed Rachid Kabbour, Fatima Gaboun, El Haj El Maadoudi, Nadia Benbrahim, Mona Taghouti, Mohammed Ouhssine, Shiv Kumar

**Affiliations:** 1National Institute of Agricultural Research (INRA), Rabat-Instituts, Rue Hafiane Cherkaoui, Agdal 10101, Morocco; younes.noutfia@inra.ma (Y.N.); mohammedrachid.kabbour@inra.ma (M.R.K.); fatima.gaboun@inra.ma (F.G.); elhaj.elmaadoudi@inra.ma (E.H.E.M.); nadia.benbrahim@inra.ma (N.B.); mouna.taghouti@inra.ma (M.T.); 2Biology Department, Faculty of Sciences-Ibn Tofail University, Kenitra 14000, Morocco; ouhssineunivit@gmail.com; 3International Center for Agricultural Research in the Dry Areas (ICARDA), Rabat 10101, Morocco; youness.ennahli@gmail.com; 4African Integrated Plant and Soil Research Group (AiPlaS), AgroBioSciences, Mohammed VI Polytechnic University, Ben Guerir 43150, Morocco; adil.elbaouchi@um6p.ma

**Keywords:** flat bread, mixture design, chickpea, milk powder, nutritional fortification, organoleptic enhancement

## Abstract

Flour quality is influenced by the nature of the gluten and its various components. Gluten free flour made of pulses is known to enhance the nutritional quality of wheat flour. However, its addition can compromise the rheological and sensorial attributes of the bread. We used mixture design to optimize nutritional and technological qualities of a wheat–chickpea flour blend by adding milk powder as a natural organoleptic improver. A total of thirteen flour blends were prepared by incorporating 10 to 30% chickpea flour and 10 to 20% milk powder to wheat flour. Our results showed that the optimal flour blend consisted of 60% wheat, 24% chickpea, and 16% milk powder. Farinographic parameters of the optimal dough blend remained on par with those of the control dough (100% wheat flour), thereby preserving its bread-making quality. Sensory analysis of breads made from the optimal flour blend revealed no significant difference (*p* ≤ 0.05) from wheat flour for crumb and chewiness. Appreciation was brought to the appearance, crust, aroma, and taste in the optimized bread. This study suggests that chickpea flour can be suitably incorporated into bread wheat flour up to a percentage of 24% with 16% milk powder to produce bread with optimal nutritional quality while improving its sensory attributes and consumer acceptability.

## 1. Introduction

Pulses are increasingly being recognized worldwide as healthy foods [1]. They are good sources of protein, vitamins, minerals, and beneficial dietary fibers for human gut health [2,3]. The World Health Organization (WHO) has recommended the frequent consumption of pulses in order to reduce serum cholesterol levels and the risks of coronary heart disease and cancer in humans [4]. This unique nutritional and health potential imparts upon pulses an important role to address issues related to the triple burden of malnutrition. Thus, the FAO of the United Nations Organization celebrated 2016 as the International Year of Pulses [5] with the aim of drawing attention to the nutritional and health benefits of pulses and strengthening the paradigm shift towards including more of this nutritional powerhouse in diets all over the world [6]. In this context, systemic attention has been given to the research of new technologies for introducing them into cereal-based foods as healthy ingredients. Pulses, especially when blended with cereals, offer a promising alternative source for nutritional and functional proteins and will carry nutritional benefit to a wide range of the population that is increasingly interested in pulse utilization [7]. The addition of pulse flour into wheat flour is a difficult task from a technological point of view because of the absence of gluten [8] as well as in relation to sensory properties, in particular, flavor and texture, which are very important in the developed world, where consumers have extremely discriminating tastes [1].

Chickpea (*Cicer arietinum* L.) is an important grain legume with a multitude of nutritional and health benefits [9]. Indeed, it contains a high proportion of non-digestible carbohydrates and contains a rich variety of phytochemicals, including natural antioxidants such as phenolic or flavonoid compounds that decrease oxidative stress and inflammation related to many chronic ailments [10,11]. The fortification of bakery products with chickpea flour has already been employed and appears to be promising in particular for the functional food market [12]. The addition of chickpea flour to wheat flour improves the quality of protein, fiber, and mineral content of the bread. Unfortunately, a threshold of 5 to 10% of chickpea flour is imposed in order to not lose some organoleptic qualities such as chewiness [13,14,15], limiting the nutritional improvement that this legume could bring to bread. This is mainly due to gluten dilution caused by the addition of chickpea flour into wheat flour. In fact, a low content or a lack of gluten may deteriorate bread quality given that gluten is a protein of major importance in baked products because of its contribution to the viscous and extensibility properties that positively influence the overall quality of bread making [16,17].

Milk powder is widely used in bread formulation due to its nutritional and technological roles [18,19]. Its incorporation into gluten free products is all the more accentuated given the distinguished quality of its ingredients, in particular, proteins and minerals ensuring a network development similar to that formed by the gluten, ensuring dough stability [20]. It also improves the mouth feel and the overall acceptability of the formulated breads [21]. For this double role, milk powder was used as a third ingredient with the objective of optimizing the nutritional and technological quality of wheat–chickpea flour blend through the mixture design methodology. Rheological and sensorial qualities of the optimized dough mixture and bread were assessed and compared to bread wheat as a control.

## 2. Materials and Methods

### 2.1. Raw Material

In this study, bread wheat (*Triticum aestivum*) of the ‘Khadija’ variety and chickpea (*Cicer arietinum*) of the ‘Farihan’ variety that is widely grown in Morocco were used for flour blend preparation. The milk powder that was used was of a known international trademark. After cleaning and the removal of foreign impurities, wheat grains were thoroughly milled in a laboratory pilot mill (BUHLER-Switzerland) with a sieve of 250 μm that was set at a 64.5% extraction rate. Chickpea grains were ground by a hammer mill (IKA, Königswinter, Germany) with a sieve of 1 mm. All of the ingredients were stored properly until needed.

### 2.2. Blend Preparation and Mixture Experimental Design

In order to optimize the flour mixture of bread wheat (A), chickpea (B), and milk powder (C) from nutritional and technological points of view, the experimental design approach that was adopted was mixture design. This approach has the advantage of rationalizing analysis cost, time, and efficiency by reducing the number of experiments and giving maximum information on the constituents, their influences taken separately, and on their eventual interactions [22]. The mixing experimental design was used during the last few years to study the formulation in many food industries for several quality purposes [23]. Regarding flour blending, it has been used for both nutritional purposes enrichment [24,25] and for improving functional quality [26,27].

Preliminary tests defined the study area by setting the lower and upper limits as 60–80% for wheat flour, 10–30% for chickpea flour, and 10–20% for milk powder. The experimental matrix (scheffe matrix) obtained was composed of 13 blend combinations in which the ingredient quantities were within the above set limits. Responses to parameters concerning the nutritional composition were proteins, crude fat, carbohydrates, total polyphenol compounds (TPC), antioxidant activity and mineral content (Fe, Zn and P), and the technological parameters were the zeleny volume and the yellow index (b*). Trace plot curves and 2D and 3D response surface presentations allowed for the visualization of the effect of variability of one ingredient on the responses while the other two ingredients were fixed.

### 2.3. Nutritional Composition Analysis

Chickpea flour, bread wheat flour, and their blends with milk powder were subjected to protein content analysis using the Kjeldahl method and crude fat analysis by diethyl ether extraction using a soxhlet apparatus. The available total carbohydrates were determined by difference calculations. All of these analyses were performed according to the official methods of analysis of the Association of Official Analytical Chemists (AOAC) [28].

#### 2.3.1. Mineral Content

Total Fe, Zn, and P concentrations in different flour blends were determined using the nitric acid–hydrogen peroxide digestion method [29,30]. Mineral concentrations were measured through inductively coupled plasma-optical emission spectroscopy (ICP-OES); (ICP-7000 Duo, Thermo Fisher Scientific, Waltham, MA, USA).

#### 2.3.2. Total Phenolic Content

The total phenolic content of flour blends was determined following the method of Singleton and Rossi [31] using the Folin–Ciocalteu phenol reagent. A total of one gram of the blend flour was extracted for 4 h with 20 mL of 80% ethanol. The mixture was centrifuged at 2000 RPM for 15 min, and 200 μL of the extract was mixed with 1 mL of Folin–Ciocalteu reagent (previously diluted 10-fold with distilled water) and was allowed to stand at room temperature for 4 min. A total of 800 μL of sodium bicarbonate (7.5 g/L) solution was then added to the mixture. After 2 h of incubation in darkness at room temperature, the absorbance was measured at 765 nm. The results were expressed as the µg Gallic Acid Equivalent (GAE) per gram of dry matter.

#### 2.3.3. Estimation of DPPH Scavenging Activity

For each flour blend extract, DPPH scavenging activity was evaluated according to the method described by Brand-Williams et al. [32]. A total of 200 μL of each blend extract at different concentrations was added to 1.8 mL of a methanolic solution of 2,2-Diphenyl-1- picrylhydrazyl (DPPH) (0.025 g/L). A negative control was prepared by mixing 50 μL of methanol with 1.95 mL of the methanolic solution containing DPPH. The absorbance of each supernatant was measured against the negative control at 517 nm after 30 min of incubation in darkness at room temperature. The estimation of the DPPH scavenging activity was calculated as follows:(1)$$\%\mathrm{Scavenging}\mathrm{Activity}=\left[\frac{\mathrm{cAbs}-\mathrm{sAbs}}{\mathrm{cAbs}}\right]\times 100$$
where c Abs represents the negative control absorbance, and s Abs represents sample absorbance.

### 2.4. Determination of Technological Quality Parameters

The yellow index (b*) of flour blend samples was measured using a Chroma Meter CR-400 (Konica Minolta-Carrieres-sur-seine, France). The instrument was calibrated against a white calibration plate. In addition, gluten strength was determined using the sedimentation volume–Zeleny volume according to the international standard ISO 5529 [33]. The results were expressed as the sedimentation volume (mL) of the flour blends in a lactic–isopropanol solution.

### 2.5. Optimal Blend

The optimal combination of the studied responses was obtained by ‘the desirability function’ as outlined by Derringer and Suich [34]. This purely mathematical procedure consists of simultaneously seeking the maximum of all the answers by the following formula:(2)$$\mathrm{Dg}(\mathrm{blend}\;)=\sqrt[{n}]{\left(d1\times d2\times d3\times \ldots \times dn\right)}$$
where Dg is the overall desirability of the flour mixture, and di refers to the individual desires of the ingredients.

### 2.6. Rheological Quality and Bread Making Procedure of the Optimal Flour Blend

The optimal flour blend was analyzed by a farinograph (Brabender farinograph© - Pfullingen, Germany) and was subjected to bread making against a control flour (100% bread wheat). Absorption, stability, and other characteristics of the dough were determined by farinograph curves according to the AACC method 54-21 [35].

### 2.7. Breadmaking

Flat breads obtained with the optimized blend and control flours were prepared and baked using the straight dough method. A total of 1 kg of flour from each sample was mixed with 2% lyophilized yeast and 1% salt. The added water volume corresponded to the maximum absorption given by the farinograph, and the kneading was realized by a spiral mixer (100–200 RPM). The dough was then fermented for 90 min at 30 °C, divided to 150 g round pieces, and was allowed to rest at 30 °C for 11 min. Each piece was manually sheeted to a round shape of a 1 to 2 cm thickness, panned, and proved for 45 min at 30 °C. The bread was baked at 240 °C for 20 min [36].

### 2.8. Sensorial Analysis of the Optimal Blend Bread

Sensory evaluation of baked breads was conducted for six defined quality attributes: appearance, chewiness, aroma, crust, crumb, and overall taste by a panel of 30 semi-trained members (employees of a catering company, researchers, and students) on a 9-point hedonic scale [37].

### 2.9. Statistical Analysis

A statistical test of the mixture design was conducted using Design Expert Software v11. The model response fitting was verified by the ANOVA test, analyzing the *f* and *p* value (*p* ≤ 0.05), the smallest standard error, the smallest mean, and the sum of the predicted deviations, and maximizing the R^2^ and the adjusted R^2^ [38]. The highest polynomial orders with significant additional terms were selected. The optimality was assessed by the D-optimal algorithm, and all ingredients were designed as dependent variables by the relation A + B + C = 1. The entirety of the experiments were conducted in triplicate, and the differences between treatment means were determined by Duncan’s multiple range test (*p* ≤ 0.05) using SPSS statistics 17.

## 3. Results and Discussion

### 3.1. Nutritional Composition of Flour Mixtures

#### 3.1.1. Macronutrient Content

The significance of proteins, carbohydrates, and lipids of the flour mixtures was investigated according to the F test and *p* value analysis, and all followed a linear model (Table 1). Factor coefficient assessment showed that protein was significantly influenced by the rate of chickpea flour and milk powder incorporation with maximum value of 15.83 g/100 g DM in the mixture consisting of 60% bread wheat, 25% chickpea, and 15% milk powder (Table 2). The lipids increased with the increase of milk powder, ranging from 4.31 to 8.98 g/100 g DM (Figure 1). In contrast, carbohydrates were more influenced by the rate of bread wheat, ranging from 63.89 to 69.99 g/100 g DM (Table 2). These findings agree with the work of Ouazib et al. [39], who demonstrated that the enhancement of the protein and lipid content of the flour blends was a direct result of the typical composition of chickpea flour that is rich in protein and lipids and that is especially rich in polyunsaturated fatty acids. Likewise, milk powder also enhanced total the protein and lipid content [40,41].

#### 3.1.2. Mineral Composition

Micronutrient response equations were not all of linear order and proved the significance of nonlinear terms. Phosphorus was the only element following a linear order equation (*p* = 0.0055). Coefficient analysis demonstrated a significant enrichment effect of chickpea flour and milk powder on phosphorus in the blend flours (Table 1) ranging from 206.4 to 6419.2 ppm. The maximum value was recorded for the mixture composed of 60% wheat, 20% chickpea, and 20% milk powder (Table 2).

The *p* value and F test showed that the equations for Fe and Zn content were of quadratic order (*p* ≤ 0.0001 for Fe; *p* = 0.0002 for Zn). Analysis of the linear terms of the Fe and Zn regression equations revealed that the iron content of a blend was more influenced by the rate of incorporated chickpea flour, while the Zn content was more influenced by the milk powder rate, demonstrating the highest concentration of 48 ppm Zn and 57.60 ppm Fe in the blend of 60% bread wheat, 20% chickpea, and 20% milk powder. The *p* value and F test of the quadratic terms of the two equations confirmed significant positive values of the partial factor (BC), which demonstrates the synergistic effect of the interaction between chickpea flour and milk powder for the maximum Fe enrichment of the mixture. Fe ranged from 33.60 to 57.60 ppm, and the maximum value was reported in the mixture of 60% bread wheat, 30% chickpea, and 10% milk powder. This interaction was insignificant for Zn, which can be visualized in the trace plot curves and response surface presentations (Figure 2).

The negative value of the partial factor AB of the iron regression equation demonstrated that the interaction between bread wheat and chickpea resulted in the minimum value of this element. The same factor being positive for Zn demonstrated a synergistic effect between bread wheat and chickpea in the increasing zinc content of the blends. The interaction between bread wheat flour and milk powder was insignificant for the iron content, while it was negative and decreased the Zn level to minimal levels.

The synergistic effect of chickpea flour and milk powder on the mineral fortification can be justified by the richness of these two ingredients in these micronutrients. Man et al. [13] reported that the addition of 30% chickpea flour increased the ash content of wheat flour by four times. An increase was observed in the same elements during bread fortification with different dairy products (milk powder, whey protein concentrate powder, and buttermilk powder) with significantly higher levels than wheat bread [41,42].

#### 3.1.3. Total Polyphenols and Scavenging Activity of Blend Preparation

The regression equation of the total polyphenolic content was of a special cubic model (*p* = 0.0045) (Table 1). By analyzing the coefficients of the linear terms, the addition of chickpea flour to the blend had the highest effect on enhancing the total polyphenol content followed by bread wheat flour, with a maximum value of 575.95 µg GAE/g DM in the mixture of 70% wheat, 20% chickpea, and 10% milk powder (Table 2). Significant quadratic terms (AC and AB) had positive coefficients demonstrating a synergetic effect of binary mixtures on enhancing the content of these bioactive molecules. The ternary mixture ABC had a negative coefficient, which might decrease this response due to dilution with milk powder. These results are in agreement with the graphic representation of this response (Figure 3).

Concerning the scavenging activity, the validated model was special quartic (*p* = 0.0057), and the scavenging activity ranged from 12.73 to 27.72%. Milk powder and chickpea flour positively influenced this response in bread wheat flour through significant terms (B, C, AB, and A²BC). This might be due to the chickpea content in terms of the polyphenols. Earlier studies undertaken on dough and pasta revealed that products made with wheat flour fortified with chickpea exhibited a higher polyphenol content and antioxidant activity, eventually doubling the control content [43]. The positive effect of milk powder on scavenging activity may be due to the natural milk antioxidants or due to the industrial supplementation of these molecules [44].

### 3.2. Technological Quality of Flour Mixture

The regression equation of the zeleny volume was of a linear order (*p* = 0.002) and was positively influenced by the addition of wheat flour. This makes sense given the nature of the added ingredients. The addition of chickpea flour and milk powder caused the dilution of the gluten provided by wheat flour (Figure 3). Bojnanská and Urminská [45] also reported a lower zeleny value with the addition of natural gluten free ingredients in wheat flour.

Concerning the yellow index, the regression equation was of a quadratic order (*p* = 0.0002) (Table 1), where the chickpea ratio had the most significant effect on increasing this parameter among the linear terms that were succeeded by wheat flour, while milk powder had a diluting effect. Binary terms (AC and BC) indicated a positive interaction on the yellow index. These results are in agreement with the results of Yıldırım and Karaboğa [46], who demonstrated that the addition of chickpea flour to bread wheat flour increased the yellow index (b*).

### 3.3. Blend Optimization

The optimization of the flour mixtures was based on the mathematical equation of Derringer and Suich [34]. This operation consisted of looking for the concomitant graphical and numerical optimum involving the addressed answers for the nutritional (proteins, lipids, carbohydrates, Fe, Zn, P, polyphenols, and scavenging activity) and technological (zeleny volume, yellow index) levels. The ingredients were set as independent factors by maximizing the level of incorporated chickpea flour and minimizing the level of bread wheat flour and milk powder (Appendix A). According to the numerical optimization, the optimal point corresponded to the mixture composed of 60% bread wheat flour, 23.8% chickpea flour, and 16.2% milk powder. The graphical optimization of the mixture revealed the same proportions obtained by the numerical optimization, corresponding to a desirability of 0.77 (Appendix A).

### 3.4. Farinograph Test

The optimization of the nutritional quality of the dough by replacing part of the bread wheat flour with chickpea flour and milk powder brought slight modifications in the farinograph results (Table 3). Water absorption did not differ between the control (bread wheat dough) and the optimal blend dough, even if the proportion of chickpea flour reached 26% in the flour mixture. These results are contrary to the results of Mohammed et al. [14], which showed that the water absorption of the dough steadily increased with addition of chickpea flour from 10 to 30%. This discordance can be explained by the potential functional interaction between the milk powder and the chickpea proteins and gluten. Indeed, by its protein and pentosan content, chickpea flour grants a high water absorption capacity to dough [47]. In parallel, milk powder has a distinct tendency for water absorption compared to wheat flour [48]. In fact, milk proteins decrease water absorption in dough [49]. These different behaviors probably affected the protein matrix of the blend when the three ingredients were together, and thus maintained the water absorption at the same level as that of the control. This parameter is considered as an important factor in commercial bread dough production, for controlling dough handling, proofing, and baking and sensorial characteristics are necessary to produce desired baked foods such as bread [50].

Regarding development time, the time required to reach the consistency of 500 BU for the optimal dough was significantly (*p* < 0.01) higher than that of the control (Table 3). This might be attributed to the milk caseins [19].

The optimal blend showed a significantly (*p* < 0.05) lower softening degree in comparison to the control dough. When chickpea was used alone, its proportion linearly increased the softening degree [14]. Dough stability remained unchanged between the control and the optimized flour doughs (Table 3). However, it decreased consistently when chickpea flour was used alone, with increased levels in wheat bread fortification due to the dilution and interference executed by the fibers provided by the chickpea flour on the gluten network [14,47,51]. The findings of Kenny et al. [19], Madenci and Bilgiçli [41], and Alsuhaibani [42] also indicated that the addition of milk whey protein concentrates, butter proteins, and milk improved dough properties in terms of stability. This parameter might also be improved by the high calcium content of milk powder, as the addition of salt has shown a positive effect on dough stability [52].

Analysis of the farinograph parameters predicted that the optimal flour blend of bread wheat, chickpea, and milk powder in a 60:24:16 ratio preserved approximately the same bread-making quality of the control wheat flour. Even if there was small difference in terms of development time, the hydration capacity and stability remained unchanged. The decreased softening degree demonstrated support to the gluten network during kneading, which is mainly due to the rheological role played by the milk powder, which has demonstrated the ability to enhance the overall physical properties of wheat pan bread [40], fighting the worsening bread-making attitude brought by chickpea on wheat flour [47].

The functional improvement brought by milk powder to the wheat and chickpea flour blend could be compared to the gluten effect on the same matrix tested by Singh et al. [51]. The addition of gluten at a 3% level was found to effectively improve the rheological characteristics of wheat–chickpea flour blends with the maximum addition level of 20%. This proved that we managed to obtain a nutritionally optimized mixture with good baking quality [14,53] without the use of gluten.

### 3.5. Sensory Profile of Optimized Bread

The sensory acceptance of the formulated breads on the basis of the scores given by the panelists (Table 4, Appendix A) showed no significant difference between the optimal and control bread with regard to crumb and chewiness. However, appearance, crust, aroma, and the taste of the bread formulated with the optimal flour blend scored higher appreciation (Appendix A).

These results suggest that in addition to the nutritional and rheological improvement brought to the flour and dough of the mixture (bread wheat–chickpea–milk powder), the bread made with the optimal blend gained the confidence and acceptability of the interviewed panel. This improvement in sensory quality may be due to the conjugated action of the milk powder and chickpea flour. The milk powder enhanced the crust and crumb softness [54] and thus corrected the negative action of the high chickpea ratio (26%) on these characteristics. Singh et al. [51] demonstrated that the softness of chapati, an Indian pancake, deteriorated with the increase of a chickpea flour level beyond 20%. In another study by Yamsaengsung et al. [55], crumb softness could not be obtained in bread made from chickpea flour, and despite the use of water and emulsifier, the latter did not correlate positively with the alveoli improvement brought by these two elements.

Due to its functional properties, milk powder upgraded dough viscoelastic development and thus improved the softness of the crumb, confirming the results obtained by Sanchez et al. [56]. Indeed, Sharma et al. [57] revealed that whey proteins had good emulsifying properties and contributed to the correct distribution of fat in the dough, thus ensuring crumb softness.

The aroma and taste of the optimal blend bread were better than those of the control bread. This may be due to the action of the chickpea flour, which, as reported earlier by Miñarro et al. [58], indicates that good sensory behavior could be achieved with the addition of chickpea flour, and it could be a substitute to soya protein in bread making.

Crust improvement was expressed by the specific color development in the optimal blend bread. The darkness and yellowness of bread increased, and this might be attributed to the milk solids, which are useful color contributors due to the Maillard reaction, the products of which positively influence the flavor of bread [59]. This reaction was also favored by the high lysine content of the chickpea powder [14].

## 4. Conclusions

The present study, conducted through the mixture design methodology, obtained an optimal nutritional mixture based on 60% bread wheat, 24% chickpea flour, and 16% milk powder. The milk powder not only boosted the nutritional potential of the mixture, but it also improved the overall bread making quality of the chickpea–bread wheat mixture in comparison to that of wheat bread by enhancing its rheological and sensorial qualities in regard to chewiness and crumb, which remained unchanged and thus overcame the technological inconvenience of the addition of chickpea flour. This opens the way for future in-depth research on the effect of milk fractions (caseins, whey, etc.,) on the nutritional and rheological quality of cereal–legume composite flours as well as the behavior of milk proteins in this matrix.

## Figures and Tables

**Figure 1 foods-10-01843-f001:**
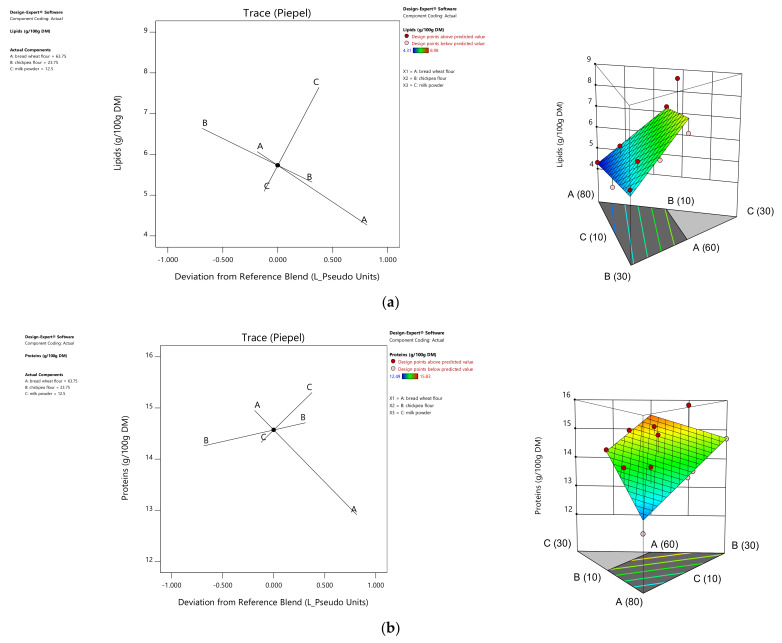
Trace plot curves and response surface (2D and 3D) presentations of lipid (**a**), protein (**b**), and total carbohydrate (**c**) variations in flour blends (A: Bread wheat, B: Chickpea, C: Milk powder).

**Figure 2 foods-10-01843-f002:**
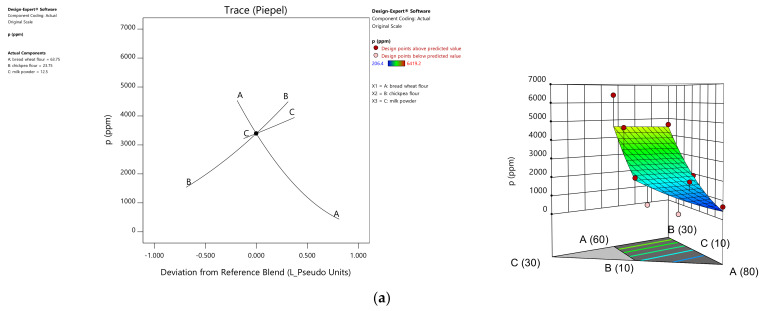
Trace plot curves and response surface (2D and 3D) presentations of phosphorus (**a**), iron (**b**), and zinc (**c**) variation in flour blends (A: Bread wheat, B: Chickpea, C: Milk powder).

**Figure 3 foods-10-01843-f003:**
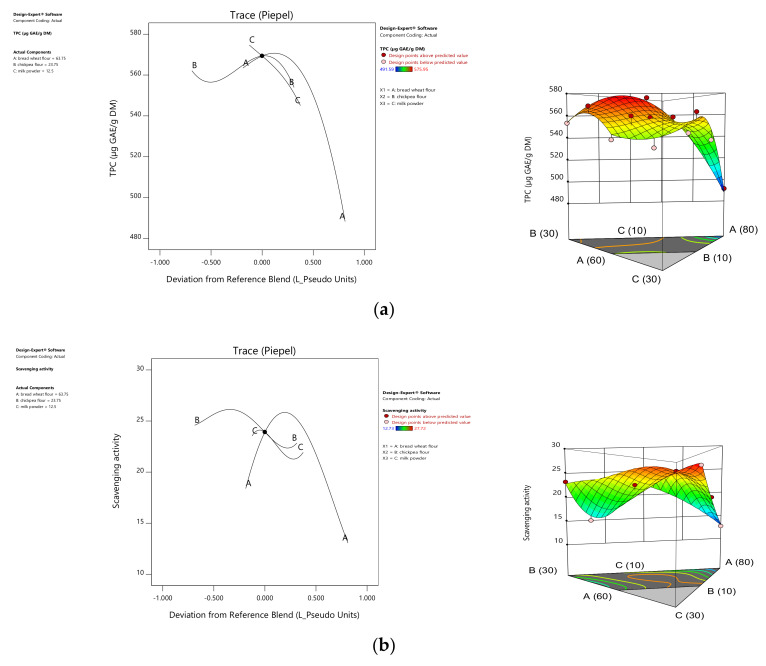
Trace plot curves and response surface (2D and 3D) presentations of total phenolic content (**a**), scavenging activity (**b**), zeleny volume (**c**), and yellow index (**d**) variation in flour blends (A: Bread Wheat, B: Chickpea, C: Milk powder).

**Table 1 foods-10-01843-t001:** Analysis of the predicted model equations for nutritional and technological quality parameters of wheat–chickpea–milk powder composite flour.

Response		Model	*p* Value	R^2^	R^2^adj	*F* Value	Prediction Equation (Pseudo Component)									
							A	B	C	AB	AC	BC	ABC	A ²BC	AB ²C	ABC ²
	Proteins	Linear	0.0078	0.6211	0.5453	8.19	12.91 *	14.71 *	16.28 *							
**Nutritional composition**	Total carbohydrates	Linear	0.0005	0.7798	0.7357	17.70	69.59 **	66.02 **	61.20 **							
	Lipids	Linear	0.0006	0.7722	0.7266	16.95	4.27 **	5.32 **	10.20 **							
	p	Linear	0.0055	0.6473	0.5767	9.18	16.74 *	65.87 *	67.82 *							
	Fe	Quadratic	0.0001	0.9593	0.8980	33.03	46.95 **	56.59 **	33.68 **	−69.30 ***	40.21	46.23 *				
	Zn	Quadratic	0.0002	0.9541	0.9213	29.08	33.42 ***	43.96 ***	52.61 ***	31.81 **	−44.17 *	−5.43				
	TPC	Special Cubic	0.0045	0.9204	0.8407	11.56	488.3 *	553.93 *	440.18 *	206.77 *	387.88 *	207.32	−721.51 *			
	Scavenging activity	Special Quartic	0.0057	0.9755	0.9264	19.87	13.06 *	22.84 *	50.12 *	16.61 *	−13.74	−66.40 *		350.83 *	269.23	−326.58
**Technological parameters**	Zeleny volume	Linear	0.0021	0.7084	0.6539	12.15	12.89 *	10.63 *	8.69 *							
	Yellow Index b *	Quadratic	0.0002	0.9507	0.9155	27.01	12.69 ***	17.90 ***	−7.05 ***	1.55	44.04 **	44.12 **				

* (*p* value ≤ 0.05); ** (*p* value ≤ 0.001); *** (*p* value ≤ 0.0001). R^2^: Coefficient of determination; R^2^adj: Adjusted coefficient of determination; *F* value: F-test value.

**Table 2 foods-10-01843-t002:** Nutritional and technological quality parameters of design mixture experiments with wheat–chickpea–milk powder composite flours.

Exp No.	Run	Lipids (g/100 g DM)	Proteins (g/100 g DM)	Total Carbohydrates (g/100 g DM)	TPC (µg GAE/g DM)	Scavenging Activity	Zn (ppm)	Fe (ppm)	P (ppm)	Zeleny Volume (mL)	b *
1	12	4.31 i	12.49 h	69.99 a	491.59 f	12.73 f	33.60 de	46.40 bc	660.80 k	13.80 a	13.02 b
2	2	7.27 b	14.63 d	65.29 efg	567.35 ab	27.72 a	32.00 de	48.00 abc	2059.20 h	10.80 ab	13.69 ab
3	3	5.59 h	14.69 d	66.55 cde	553.48 cd	23.21 bcd	44.80 ab	57.60 a	4630.40 c	10.30 ab	18.11 ab
4	8	7.10 c	15.17 b	64.64 fg	544.34 de	20.43 d	48.00 a	57.60 a	6419.20 a	9.30 b	16.42 ab
5	7	5.59 h	14.26 e	67.03 cd	540.18 e	20.20 d	30.40 e	52.80 ab	206.40 m	11.30 ab	15.53 ab
6	4	4.38 i	13.76 f	68.93 ab	575.95 a	22.07 cd	46.40 a	33.60 d	1889.60 i	11.30 ab	15.00 ab
7	13	8.98 a	15.11 bc	63.89 g	537.07 e	24.17 abcd	40.00 bc	51.20 abc	4636.80 b	10.55 ab	15.13 ab
8	5	6.37 e	15.83 a	64.50 fg	570.81 ab	16.25 e	43.20 ab	57.60 a	2646.40 f	11.55 ab	19.11 a
9	11	6.11 f	15.04 c	65.58 defg	560.44 bc	25.11 abc	44.80 ab	46.40 bc	2262.40 g	10.55 ab	18.23 ab
10	10	5.62 h	14.24 e	67.04 cd	544.99 de	23.49 bcd	36.80 cd	43.20 cd	1763.20 j	11.80 ab	16.24 ab
11	6	5.79 g	13.43 g	67.36 bc	561.85 bc	26.22 ab	36.80 cd	48.00 abc	459.20 l	10.80 ab	17.24 ab
12	9	6.09 f	13.71 f	66.14 cdef	560.96 bc	23.89 abcd	44.80 ab	48.00 abc	3299.20 e	10.55 ab	19.50 a
13	1	6.68 d	15.20 b	64.51 fg	563.01 abc	23.67 abcd	44.80 ab	49.60 abc	3372.80 d	10.30 ab	18.50 ab

Means with different letters within the same column are significantly different according to Duncan’s test (*p* < 0.05). DM: Dry Matter; GAE: Gallic Acid Equivalent.

**Table 3 foods-10-01843-t003:** Farinograph parameters of the optimal composite flour dough.

Treatments	Dough Development Time	Consistancy	Water Absorption	Stability	Softening Degree
	mn:ss	BU	%	mn:ss	BU
Control(bread wheat dough)	1:07 a	553	51.8	1:09	82 a
Optimized blend dough	1:55 b	527	59.0	1:44	71 b
Probability	0.01	ns	ns	ns	0.05

Values with different letters are significantly different at *p* ≤ 0.01 or *p* ≤ 0.05; ns are not significantly different. BU: Brabender Unit; mn: minute; ss: seconds.

**Table 4 foods-10-01843-t004:** Sensory quality scores of optimized blend bread compared to wheat bread (control).

	Sensory Quality Scores					
	Appearance	Crust	Crumb	Chewiness	Aroma	Taste
Optimized blend bread	7.6 a	7.8 a	6.9	7.2	7.5 a	7.7 a
Control (100%Bread wheat)	6.1 b	6.7 b	7.0	7.3	6.2 b	6.4 b
Probability	0.01	0.01	ns	ns	0.05	0.01

Values with a different letter within the same column are significantly different at *p* ≤ 0.01 or *p* ≤ 0.05 as indicated; ns values are not significantly different. Each value is an average of 30 replications.

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
