# Peer review of "Nutritional and Technological Optimization of Wheat-Chickpea- Milk Powder Composite Flour and Its Impact on Rheological and Sensorial Properties of Leavened Flat Bread"

_foods, 2021, doi:10.3390/foods10081843_

Round 1
Reviewer 1 Report
Dear Authors,
the manuscript was somehow improved, but in a very limited degree. The main drawbacks that neither the volume nor texture was analyzed still exsist. The addition of farinograph measurement for only two samples (control and optimized blend) is not enough to overcome the mentioned drawbacks. Furthermore, statistics for Table 3 is presented incorrectly. Levels of p≤0.01 or p≤0.05 are thresholds for (high) significance, but the excat p levels for each test have to be listed.
Reviewer 2 Report
the manuscript has been significantly improved.
Author Response
Thank you very much for your encouragements
Reviewer 3 Report
The authors have developed wheat-chickpea-milk dough rich in nutrients and with high bread-making quality. The quality of the bread was evaluated by semi-trained 30 panels to be equivalent to the conventional wheat bread.
The interesting point of the research is its strategic approach to determine the composition of the optimal flour blend. The experiments have been performed professionally with clear demonstration of the results. The manuscript is well-organized and the logic flow of the discussion is easy to follow.
Minor points:
- Cicer arietinum (Line 59) > Italic
- ..by Brand-Williams [27]. (Line 138) > ..by Brand-Williams et al. [27].
- "with good backing" (Line 379) > "with good baking"
- The reference list should be re-examined:
References #28:
International Organization for Standardization. ISO 5529:2007 Wheat—Determination of the Sedimentation Index—Zeleny Test;
International Organization for Standardization: Geneva, Switzerland, 2007.
References #30:
AACC. Approved Methods of the American Association of Cereal Chemists, 10th ed.; Cereals & Grains Assn: St. Paul, MN, USA, 2000.
References #41:
- and ... > Yıldırım, A. and ...
References #46:
SINGH,... > Singh,...
Best regards,
Reviewer
Reviewer 4 Report
- The title covers the main aspects in this paper, it reflects the aim and scientific purpose of the experiment conducted.
- The abstract explains the meaning of the paper and includes the background, results and conclusion parts.
- The introduction provides a good generalized background of the topic, giving the reader an idea of the wide range of applications of this technology. Suggestion to the authors to shorten the text from lines 37-73 in the introductory part of the manuscript. It seems a little disjointed that way.
- The methods used in this paper are appropriate to the aim of the study. The methods are clear and replicable.
- it would be good to write information about the number of replicates of the performed analyzes;
- in some places the reference of the performed method is missing;
- In the chapter Materials and Methods it is necessary to give more information about the Mixture Experimental Design
- Why are other color parameters not shown (L * and a *)?
- All mathematical formulas must be written with an appropriate tool (MS Equation) and numbered. All variables in the mathematical formula must be clarified and stated below;
- improve the quality of figures presented;
- The conclusions presented in this paper are consistent with the results found. Given the volume of results presented, there is a need to improve the conclusions section: focus more on how your research contributed to knowledge gaps and describe research limitations for future research.
Round 2
Reviewer 1 Report
Dear Authors,
as already mentioned, the manuscript was somehow improved, but in a very limited degree. The main drawbacks that neither the volume nor texture was analyzed still exsist. The addition of farinograph measurement for only two samples (control and optimized blend) is not enough to overcome the mentioned drawbacks.
If it is not possible to include volume as most important quality parameter for research or to perform farinograph measurements for all samples, I strongly recommend to add extensograph and sedimentation measurements of control bread and optimized blend. This would enlarge scientific quality and the mentioned additional measurements are completed within one day only.
Author Response
Dear reviewer,
First of all, we thank you for your relevant remarks and we are convinced that the realization of your suggestions will further enrich the scientific quality of this work. Unfortunately we do not have an extensograph in our laboratory so we will not be able to do this analysis quickly.
For the sedimentation analyzes, we have carried out during this study the determination of the zeleny value which correspond to sedimentation volume according to the standard method mentioned in the article.
This manuscript is a resubmission of an earlier submission. The following is a list of the peer review reports and author responses from that submission.
Round 1
Reviewer 1 Report
I have read the manuscript "Nutritional and technological optimization of wheat-chickpea- milk powder composite flour and its impact on rheological and sensorial properties of leavened flat bread."
The authors have presented and discussed the different blends of Chickpea flour/wheat flour/powder milk to optimise the incorporation of chickpea flour into lat bread composition. I think the study has been well conducted.
the major hurdle for me is the complexity of understanding the 3D representation. It is very hard to read and to extract any information out if it.I would suggest the authors to use a more classical presentation approach.
I would recommend the authors to clarify the meaning and overall goal of the study. Why addition of Milk powder is important? See the minor comments?
In addition the Farinograph study has been done in comparison to a control dough. What is the control dough? Similar comments can also be made for the sensory experiment. If addition of Milk poweder is to increase the organoleptic, why the sensory test made with a basic wheat flour dough and not an appropriate Chickpea-wheat flour blend?
I have some minor concerns:
- I'm old school but why using a predictive model to define the different composition of Milk powder/ chickpea and wheat flour? I would have liked also to see wheat flour alone, wheat flour and Chickpea flour...
- What is the rational of using Milk powder? please explain.
- The introduction mentioned that the main issue with the addition of Chickpea flour into bread flour is the gluten. I suggest the authors to explain a bit further how important gluten is in the context of flat bread.
- As a follow up, how does milk powder solving the issue?
Reviewer 2 Report
Dear Authors,
the nutritional value of cereal products are of great interest. Although your manuscript covers this topic, the aim of your work is not clear. The introduction is very short and lacks of relevant information. The volume of breads as most improtant technological quality parameter is missing, furthermore texture analysis of breads were not performed. Nice figures were prepared, but again the sense behind is missing. Usually, ingredients like milk powder are used for glutenfree recipes, but not for wheat bread, except sweet bakery prodcuts. This results in loss of vegan status, which is untypical for wheat breads. One benefit could be a reduced allergenic potential, but milk proteins are allergens as well. Language quality has to be imrpoved and a kind of structure is completely missing. However, this manuscripts lacks in objectives and research questions.
Reviewer 3 Report
The manuscript “Nutritional and technological optimization of wheat-chickpea-milk powder composite flour and its impact on rheological and sensorial properties of leavened flat bread” is overall well written, the results are clearly presented with appropriate figures and tables. The text deals with an increasingly popular topic concerning the partial replacement of wheat flour with alternative raw materials, including gluten-free ones. Therefore, as also specified in the conclusions of the text, it can be a starting point to explore the topic from other points of view, fully investigating the technological properties of such ingredients. The statistical analysis of the data is appropriate and well organised even if the choice of Duncan’s test to separate the differences between the averages seems to me not very strict.
I would like to suggest the following:
Lines 55-56. To make it clearer, I would add two commas as follows:
…of the absence of gluten [8] and also in relation to sensory properties, in particular flavor and texture, which are very important in the developed world,…
Line 70. I would suggest to implement the introduction section at this point adding more information about milk powder as integral part of your study with wheat and chickpea flours.
Line 87. The word ‘order’ in the sentence must begin with a lowercase letter.
Line 106. The word ‘kjeldhal’ should be written like this ‘Kjeldahl’ and with a capital letter.
Line 187. I would suggest writing as follows: The entirety of the experiments…
Line 200. The data given in the text for carbohydrates may refer to Table 2, not Figure 1.
Line 256. I think the sentence is incomplete and there is some confusion as it mentions iron first and then zinc. Check if any concepts are missing in the sentence about zinc.
Line 293. No comments on AC interaction on yellow index. Looking at Table 1, it seems also positive despite the diluting effect.
Lines 319-320. This sentence could lead to misunderstandings, it would be more appropriate to write that ‘part of the wheat flour has been replaced’.
Line 335. Using the word ‘regarding’ instead of ‘as’ at the beginning of the sentence sounds better to the reader in my opinion.
Lines 346-347. The meaning of the sentence regarding the use of chickpea alone is not very clear. Perhaps it was meant that its concentration increased linearly the softening degree as stated in the reference [14]?
Line 375. In Table 4, in the first column, it states ‘Bread wheat bread’. Did you mean ‘Wheat bread control’?
Line 379. I woul suggest to use ‘by the mixture’ instead of ‘of the mixture’ in the sentence.
Line 384. I would specify what chapati is, not everyone knows this type of bread.
Line 400. It would be better to write ‘maillard’ with a capital letter.
